# Experimental Solid–Liquid Mass Transfer around Free-Moving Particles in an Air-Lift Membrane Bioreactor with Optical Techniques

**Naila Bouayed** [1]**, Manon Montaner** [1]**, Claude Le Men** [1]**, Johanne Teychené** [1]**, Christine Lafforgue** [1]**, Nicolas Dietrich** [1,*] **, Chung-Hak Lee** [2] **and Christelle Guigui** [1]

1   Toulouse Biotechnology Institute (TBI), Université de Toulouse, CNRS, INRAE, INSA, 31077 Toulouse, France
2   School of Chemical and Biological Engineering, Seoul National University, Seoul 151-744, Korea
*   Correspondence: nicolas.dietrich@insa-toulouse.fr

**Abstract:** This article focuses on the study of the mass transfer involved in the application of a bacterial antifouling technique for membrane bioreactors (MBR), via the addition of solid media. These alginate objects can contain a biological system capable of producing an enzyme that degrades the signal molecules responsible for membrane fouling. The objective of this article is to quantify the mass transfer by distinguishing two main types: the transfer from the liquid to the solid media and the transfer from solid media to the liquid phase. For this purpose, a model molecule was chosen, and experiments were specifically developed with an optical device to track the concentration of the dye in the liquid phase, considering three different shapes for the particles (beads, hollow cylinders, and flat sheets). The experiments were first performed in jar tests and then in a lab-scale reactor. The results of this study revealed that the total amount of dye transferred into the sheets was greater than that transferred into the cylinders or the beads, which was attributed to the sheets having a larger exchange area for the same volume. When the dyed media were implemented in the MBR (loading rate of solid media: 0.45% *v/v*—no biomass), the global transfer coefficient from the sheets to the liquid was found to be greater than for the other shapes, indicating a faster transfer phenomenon. The effect of aeration in the MBR was investigated and an optimal air flowrate for fostering the transfer was found, based on the highest transfer coefficient that was obtained. This study provided key information about mass transfer in MBRs and how it is affected by the particle shapes and the MBR operating conditions.

**Keywords:** membrane bioreactors (MBRs); air-lift reactors (ALRs); mass transfer

## 1. Introduction

A membrane bioreactor (MBR) is a wastewater treatment that combines a membrane process (microfiltration, ultrafiltration, etc.) with a biological treatment. Over the last few decades, MBRs have proved to be highly efficient for advanced wastewater treatment and reuse, and have provided high standards of effluent quality, biomass retention, biomass concentrations, organic removal efficiency, and organic loading rate, together with a low production of sludge and a small environmental footprint [1,2]. Yet the extensive development of MBRs for wastewater treatment is still restricted by membrane fouling, which results from the formation of a fouling layer on the membrane surface [3] (deposition and accumulation of solids, biofilm, pore clogging, and adsorption of products, etc.). Membrane fouling gives rise to an overall reduction in performance by causing a loss of permeability, thus increasing the energy consumption and the operating costs. Several cleaning methods (physical, chemical, and biological) have been developed in attempts to mitigate this phenomenon. One of the most commonly applied methods is air-sparging, which consists of the injection of bubbles to induce shear stress between two flat membranes, creating an air-lift reactor (ALR) [4–8]. The addition of solid particles—such as synthetic micro-particles,

granular activated carbon [9], or other granular scouring agents (reviewed by [10])—have also demonstrated an effective reduction in fouling in MBRs, due to a mechanical washing effect. Recently, a novel biological method has been developed for fouling mitigation, related to the perturbation of cell-to-cell communication between bacteria [11–15]. The technique, named quorum quenching (QQ) [16], is based on an enzymatic degradation of the signal molecules (AHLs), which are produced for bacterial communication (quorum sensing) [17], by the excreted or intracellular enzymes produced by specific QQ bacteria. Recent works have shown that when QQ bacteria are immobilized in small capsules and implemented in MBRs, the biofouling phenomenon is considerably inhibited [18]. For more information on the QQ mechanism and its application to MBRs, the reader is invited to consult the reviews in [19–22]. With the development of QQ entrapping methods [21], the mass transfer appeared as one question that it was essential to address, for two main reasons. Firstly, as in every (bio)chemical reaction, the mass transfer of the reagents is a key phenomenon as it can be a limiting step. Secondly, the importance of studying the mass transfer involved in QQ is also linked to the different entrapping methods [22–24] that have been developed (introduction of solid beads, hollow cylinders, or sheets into the reactors [25] and to the existing kinds of QQ-bacteria-producing endoenzyme or exoenzyme [25–28]. From these specificities, two cases of mass transfer can be distinguished: from the mixed liquor towards the inner part of solid media—in the case of endoenzyme degradation—and from the solid media to the mixed liquor—in the case of exoenzyme-mediated QQ. Even though some attempts have been reported in the literature previously cited, the mass transfer phenomena involved in QQ are not fully understood yet and the present study aimed to improve this situation. These studies have mostly focused on the observation of diffusion (with no liquid flow), at the scale of one single QQ medium.

The experimental approach described here was based on the use of a dye as a model molecule in order to quantify the mass transfer at the reactor scale (not only at the QQ media scale). More specifically, the objective was to provide a quantitative characterization of the mass transfer of the molecules involved in QQ, from the liquid to the QQ media, and vice versa, for different MBR operating parameters (aeration flow rate, type of QQ media, etc.). In this context, this article offers the determination of mass transfer parameters such as the transferred flux, the mass transfer coefficient, and the Sherwood number [29–31]. On the basis of these quantitative parameters, the effect of the operating conditions on the mass transfer phenomena are discussed, with reference to the effect of the shape of the QQ media and provides valuable suggestions for future research in this area.

## 2. Materials and Methods

### 2.1. Physical Properties of QQ Media

The QQ media used in this study were made of sodium alginate. Three different shapes of media were used in this study, as follows: QQ beads, QQ hollow cylinders, and QQ sheets (Figure 1).

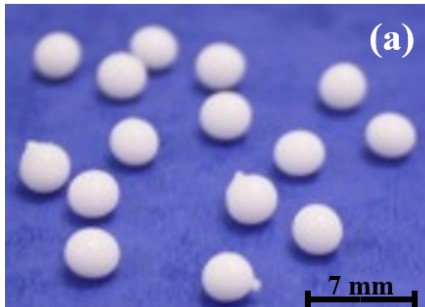 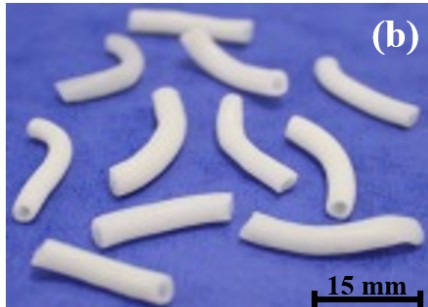 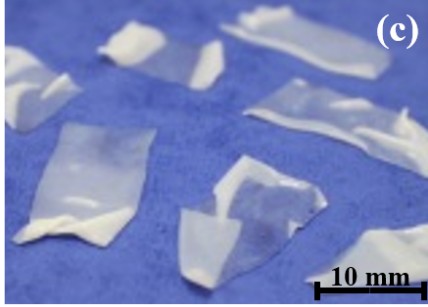

**Figure 1.** Photographs of media used in the study: (**a**) beads, (**b**) hollow cylinders, and (**c**) sheets.

The physical properties of the QQ media are summarized in Table 1.

**Table 1.** Properties of the solid media.

| Solid Medium | Beads | Hollow Cylinders | Sheets |
|---|---|---|---|
| Dimensions (mm) | Diameter: 3.5 | Inner diameter: 1.7<br>Outer diameter: 3.5<br>Length: 27 | Length: 20<br>Width: 10<br>Thickness: 0.5 |
| Volume of a particle (mm$^3$) | 22.5 | 198.5 | 100 |
| Surface area of a particle (mm$^2$) | 8.5 | 455.8 | 400 |

### 2.2. Mass Transfer Experiments

The mass transfer of real signal molecules (such as N-acyl-l-homoserine lactones, acyl-HSLs, or AHLs) was initially considered in this study. However, depending on the considered bacteria, these molecules can have various compositions and the analysis of the transfer would have required the detection and the quantification of several AHLs. Among the proposed methods, a biological one that relies on the use of the reporter strain, *A. tumefaciens,* is often proposed to detect the AHLs [32,33]. However, this method requires perfectly sterile conditions to be maintained, which could not be achieved at the reactor scale. In addition, the amount of AHLs in the mixed liquor is very small, given that these signal molecules are usually produced at very low concentrations (in the range of picograms to nanograms per liter) and that they are present as a complex mixture with different compounds, an extraction procedure is necessary before their quantification. Several methods to measure the AHL concentration after their extraction have been reported to date and are summarized in [19]. After extraction, AHL's detection by mass-spectrometry (HPLC-MS; GC-MS, LC-MS-MS; UPLC6FTIR-MS; and CZE-MS), NMR, or IR can be performed. However, the choice was made to study mass transfer at the global scale, rather than at the media scale. Thus, the selection of a mimic molecule (tracer) was preferred. Rose Bengal lactone is a light-red to pink powder used as a dye, the main properties of which are presented in Table 2. Like the AHLs, Rose Bengal lactone contains a lactone group—the reason why it was chosen as a model molecule to study the mass transfer.

**Table 2.** Physicochemical properties of Rose Bengal lactone.

| | |
|---|---|
| Chemical formula/Molecular structure | $C_{20}H_4Cl_4I_4O_5$  |
| Molecular weight (g.mol$^{-1}$) | 973.67 |
| Appearance | Pink powder |
| Purity | ≥95% |

The Rose Bengal lactone was purchased from Sigma-Aldrich (St. Louis, MO, USA). Solutions at a concentration of 0.4 g·L$^{-1}$ were prepared in tap water. In order to help the dissolution of the powder, the solutions were sonicated 3 to 4 times for 15 min, and shaken between two sonication cycles, until a clear, dark-pink solution was obtained.

The concentration of the dye was determined using optical methods. The absorbance of dye solutions was measured using a spectrophotometer (Jasco-V630, Pfungstadt, Germany). The measurements were taken at 548 nm, wavelength corresponding to the maximum absorbance in the visible range.

### 2.2.1. Mass Transfer from the Liquid to the Media

The study of the mass transfer from the liquid phase to the solid medium was performed in jar tests, in order to control hydrodynamic conditions in the different 1L-beaker reactors (Floculateur, Bioblock Scientific). QQ media were placed in a solution of Rose Bengal lactone (approximately 0.4 g·L$^{-1}$), at a solid:liquid volume ratio of 1:9, and under stirring (90 rpm) at room temperature. Two controls without QQ media (one with water and one with the saturated solution of Rose Bengal lactone) were conducted to ensure that the observed decrease in concentration during the liquid phase could only be attributed to a transfer into the QQ media, and not to additional phenomena such as natural degradation, precipitation, etc. Liquid samples were collected from the jars every 30 min at first, and then spaced out after the first 7 h. The experiments lasted 24 h—time for the concentration to reach a constant value, corresponding to the stabilization of the system. After 24 h, the stained QQ media were collected and drained, to be used for the subsequent experiments.

### 2.2.2. Mass Transfer from the Media to the Liquid

The mass transfer from the media to the liquid phase was studied in the lab-scale air-lift-membrane bioreactor (ALMBR), previously described. The saturated (stained) QQ media collected at the end of the jar-test experiments were introduced in a volume fraction of 0.45% *v/v*, with respect to the total volume of reactor (13 L). The air-lift was set to a riser width of 15 mm. Different operating conditions were investigated in order to highlight the effect of the QQ media shape, as well as the effect of the hydrodynamics (three air flowrates) on the transfer from the QQ media to the liquid phase. Three air flowrates were injected at the bottom of the reactor, corresponding to SAD (specific aeration demand) of 0.75, 0.9, and 1 Nm$^3$.h$^{-1}$.m$^{-2}$. In order to visualize the transfer phenomena in the reactor, a camera technique was developed to measure the absorbance (and, thus, the concentration of Rose Bengal lactone in the liquid phase) and used a 12-bit (4096 gray level) CMOS camera (Basler-Ace Aca1920-155 um), equipped with a green filter (495 to 505 nm). A backlight panel (Phlox-LedW-BL, 400 × 200 mm$^2$, 24 V, 2A, Phlox) was set behind the reactor, as presented in Figure 2. The observation window was 97.5 mm wide and 150 mm long and was located at the bottom of the reactor (under the membrane) so that the whole depth of the tank could be observed (thanks to the clear baffles). The camera technique was preferred to a simple absorbance measurement via a light sensor because of the coexistence of three phases in the reactor: gas, liquid, and solid, which can be taken into account more accurately, thanks to image processing.

An image was recorded every 10 s (acquisition frequency of 0.1 Hz) and the total acquisition time was 1 day or more, depending on the time required to reach equilibrium. The size of the images was 1000 × 650 pixels$^2$ (1 pixel = 150 μm). Pylon software (Basler, Germany) was used to control the camera and to configure all of its settings.

The filter was used to narrow down the range of wavelengths passing through the liquid in the reactor to approach a monochromatic beam (∼500 nm, for which the absorbance was high enough for measurement accuracy) and to link the intensity of light to the concentration of Rose Bengal lactone, according to the Beer–Lambert law (Equation (1)), where *A* is the instantaneous absorbance, *I* is the instantaneous light intensity, *I$_o$* is the light intensity through the blank (water), *C* is the concentration of Rose Bengal lactone, *l* is the optical path length, and $\varepsilon_\lambda$ is the molar attenuation coefficient. The latter coefficient was obtained from the calibration.

$$A = -\log\left(\frac{I}{I_0}\right) = \varepsilon_\lambda . l . C \qquad (1)$$

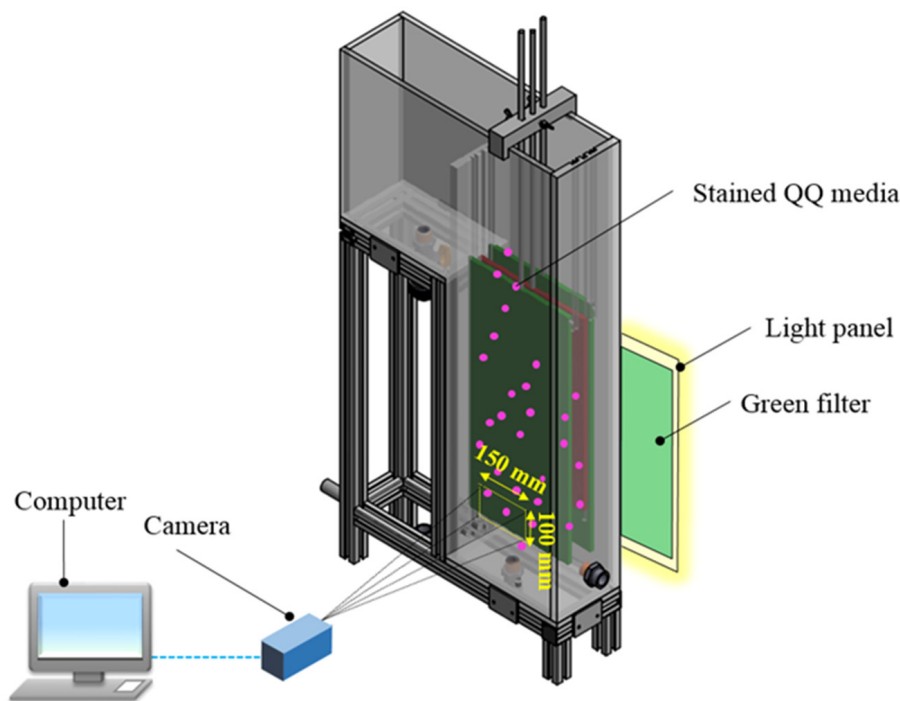

**Figure 2.** Experimental setup for the visualization of the mass transfer in the MBR.

Images of standard solutions of Rose Bengal lactone, with known concentrations (Table 2), were recorded using the camera setup, and the absorbance of these solutions was also measured using the spectrophotometer, at 500 nm (Table 3).

**Table 3.** Spectrophotometer and camera-deduced optical parameters for standard solutions of Rose Bengal lactone.

| Concentration of Dye Lactone Solution $C$ (g·L$^{-1}$) | 0.0002 | 0.0006 | 0.0012 | 0.0023 | 0.0044 |
|---|---|---|---|---|---|
| Spectrophotometer | | | | | |
| Absorbance at 500 nm $A_{spectro}$ (-) | 0.0037 | 0.0088 | 0.0188 | 0.0346 | 0.0673 |
| Optical path length $l_{spectro}$ (cm) | | | 1.0 | | |
| $\frac{A_{spectro}}{l_{spectro}}$ (cm$^{-1}$) | 0.0037 | 0.0088 | 0.0188 | 0.0346 | 0.0673 |
| $\varepsilon_{500\ nm}$ (L·g$^{-1}$·cm$^{-1}$) | | | 15.136 | | |
| Camera | | | | | |
| Absorbance at 500 nm $A_{camera}$ (-) | 0.0490 | 0.1000 | 0.2050 | 0.4180 | 0.7960 |
| Optical path length $l_{camera}$ (cm) | | | 11.6 | | |
| $\frac{A_{camera}}{l_{camera}}$ (cm$^{-1}$) | 0.0042 | 0.0086 | 0.0177 | 0.0360 | 0.0686 |
| $\varepsilon_{500\ nm}$ (L·g$^{-1}$·cm$^{-1}$) | | | 15.425 | | |

The absorbances per unit of optical path length are plotted on the same graph for comparison, in Figure 3. It can be observed that the ratios of the absorbance to the optical path length are very close for both measurement techniques. Furthermore, the attenuation coefficients, $\varepsilon_{500\ nm}$, were found to be very close (Table 2), with less than 2% deviation. From this comparison, it is possible to conclude that the camera technique, which was specifically developed for the study of the mass transfer in the ALMBR, indeed, allowed the measurement of an absorbance that followed the Beer–Lambert law. Given these results, the camera was selected as the only measurement technique to study the dye transfer from the QQ media to the liquid phase in the ALMBR.

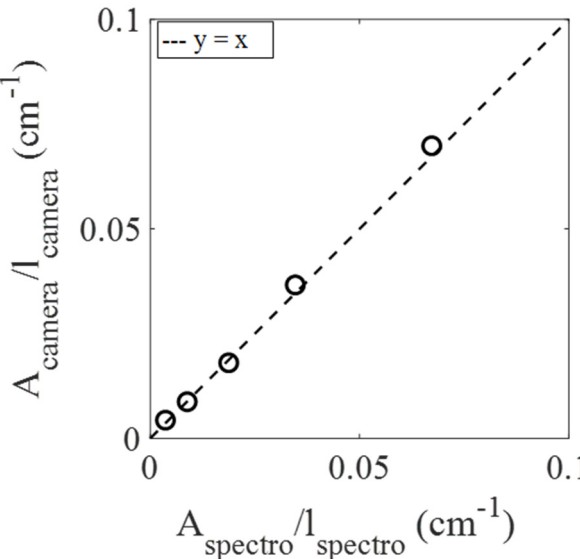

**Figure 3.** Comparison of the ratios of the absorbance to the optical path length obtained by the spectrophotometer and the camera technique.

The images recorded during the experiments were processed using a Matlab (Math-Works, Portola Valley, CA, USA) program. Prior to each experiment (before the addition of the stained solid medium), approximately 60 images of the blank (tap water) in the reactor were recorded to determine the intensity, $I_o$, and 100 images were recorded of the dark (turning the light panel off) for further calibration of the sensitivity of the measurement. After defining the total number of images to be processed, the acquisition period was selected and a threshold factor of 60% (used to remove the bubbles or solid media from the image) was applied.

The matrices corresponding to the dark images were first averaged. This average dark image was subtracted from the images to be processed in order to attenuate the signal noise (especially at low intensities). The resulting matrices of the blank gave "flat" matrices, which were averaged in a single, flat matrix. In the same way, the raw images of the experiment were processed by subtracting the matrix of the average dark image. In the following step, the threshold factor was applied in order to remove the objects (bubbles and solid particles) and only keep the pixels corresponding to the liquid phase (intensity > 60% of the maximum intensity). The mean intensity, $I$, of the processed image was then determined by averaging the remaining pixels and the intensity, $I_0$, was obtained by averaging the intensities of the corresponding pixels (in the same location) on the flat matrix. Finally, the corresponding absorbance, $A$, was determined according to Equation (1).

First of all, the stability of the Rose Bengal lactone over time was evaluated by introducing a solution of known concentration into the reactor, and by using the camera to measure the light intensity of the light panel through this solution over time. The dye was found to be sensitive to strong, long-term exposure to the light panel, as the intensity recorded increased (concentration decreased) over time. Thus, the strength of the light panel was adjusted with a view to avoiding the degradation of Rose Bengal lactone during the experiment so as to be able to draw reliable conclusions on the mass transfer.

## 3. Liquid–Solid Mass Transfer from the Liquid to the Solid Medium

### 3.1. Analysis of the Adsorption Kinetics

The main mathematical models that have been proposed in the literature to describe the overall adsorption phenomenon are based on the quantity, $q(t)$, of solute adsorbed per unit mass of solid (in $mg \cdot g^{-1}$). The pseudo-first-order model was proposed by Lagergren in 1898 [34] and is expressed according to Equation (1), where $q_e$ is the adsorbed quantity reached per unit of mass at equilibrium, and $k_1$ is the constant of the pseudo-first-order

kinetics (in $s^{-1}$). Determining this constant requires the linearization of Equation (2), which results in Equation (3), where $k_1$ is the slope of the curve, $\ln(q_e - q(t))$, versus time.

$$\frac{dq(t)}{dt} = k_1(q_e - q(t)) \tag{2}$$

$$\ln(q_e - q(t)) = \ln(q_e) - k_1 t \tag{3}$$

Another model, the pseudo-second-order model [35], assumes the existence of strong chemical bonds between the solute molecule and the solid. This model is described by Equation (4), where $k_2$ is the constant of the pseudo-second-order kinetics (in $g \cdot mg^{-1} \cdot s^{-1}$). The linear form presented in Equation (5) allows the constant, $k_2$, as well as the adsorbed quantity at equilibrium, $q_e$, to be determined.

$$\frac{dq(t)}{dt} = k_2(q_e - q(t))^2 \tag{4}$$

$$\frac{t}{q(t)} = \frac{t}{q_e} + \frac{1}{k_2 q_e^2} \tag{5}$$

In the present case, the adsorbed quantity at time—$t$, $q(t)$—can be deduced from Equation (6), where $M_S$ is the mass of solid particles; $\rho_s$ and $V_s$ are the wet density (in $kg.m^{-3}$) and the total volume (in $m^3$), respectively, of the solid particles; $C_l(t)$, is the concentration in the bulk; and $V_l$ is the total liquid volume.

$$q(t) = \frac{C_l(t=0) - C_l(t)}{M_s} V_l = \frac{C_l(t=0) - C_l(t)}{\rho_s V_s} V_l \tag{6}$$

The jar-test experiments were repeated three times. The results are presented in Figure 4, in terms of the total amount of Rose Bengal lactone per unit of the mass of the medium ($q(t)$) (defined in Equation (6)) for the different shapes of the QQ media.

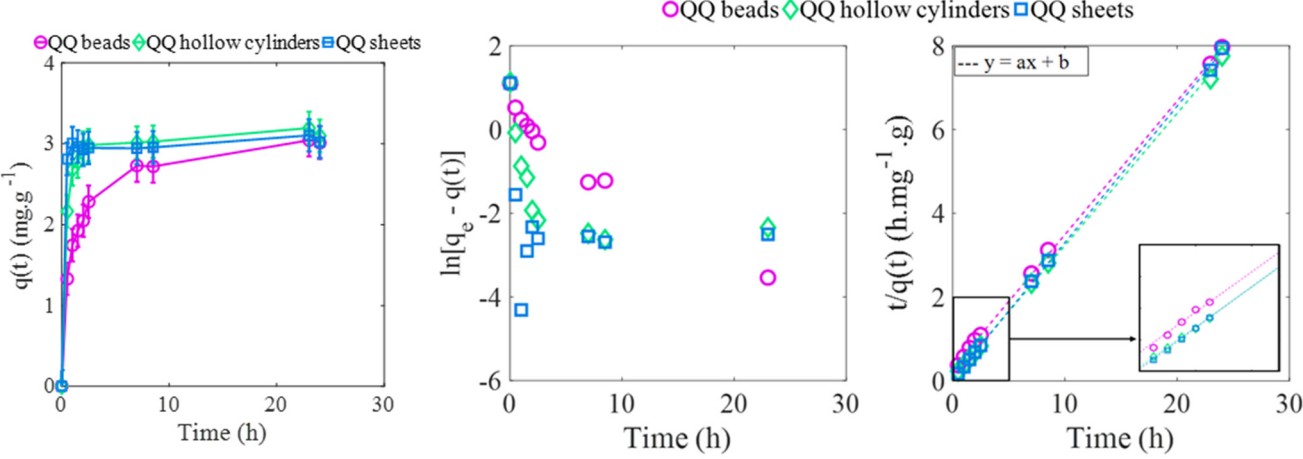

**Figure 4.** **Left**: amount of Rose Bengal lactone transferred in the solid media over time. **Middle** and **Right**: identification of the adsorption model parameters for the experimental data. Jar-test experiments.

The curves in Figure 4 describe a very rapid increase in the first hour and then an overall slowdown until equilibrium was reached, after 24 h. This type of trend is typical of a mass transfer phenomenon, in which the gradient of concentrations tends to decrease over time, thus slowing down the increase in the total amount transferred. A comparison of the three experiments performed for each medium reveals that the jar-test experiment was repeatable, with less than 10% standard deviation between the three final values of the

amount of Rose Bengal lactone that were transferred, and a similar time evolution for the experiments conducted with the same QQ medium.

The comparison of the three curves obtained for three media shapes, in Figure 5, suggests that the global adsorption dynamics depend on the shapes of the QQ media. In addition, the time to reach 96% of the final value was 0.5 h and 2 h for the sheets and hollow cylinders, respectively, whereas for the beads, this time was much longer—approximately 8.5 h. According to the adsorption theory previously presented, the pseudo-first- and pseudo-second-order models were considered to describe the kinetics of the phenomenon under study. The variations of $ln[q_e - q(t)]$ and $\frac{t}{Q}$ over time were, thus, deduced from the experimental data and are presented in Figure 4. The variations of $ln[q_e - q(t)]$ over time did not show a linear trend, unlike the variations of $\frac{t}{Q}$, which suggests that the pseudo-second-order model could be more appropriate ($R^2 > 0.99$) in describing the adsorption phenomenon. The parameters of the linear correlations ($y = ax + b$) are presented in Table 4. The parameters of the pseudo-second-order adsorption model ($k_2$ and $q_e$) were deduced from the linear correlation parameters $a$ and $b$ (from Equation (5)), according to Equations (7) and (8).

$$q_e(calculated) = \frac{1}{a} \tag{7}$$

$$k_2 = \frac{a^2}{b} \tag{8}$$

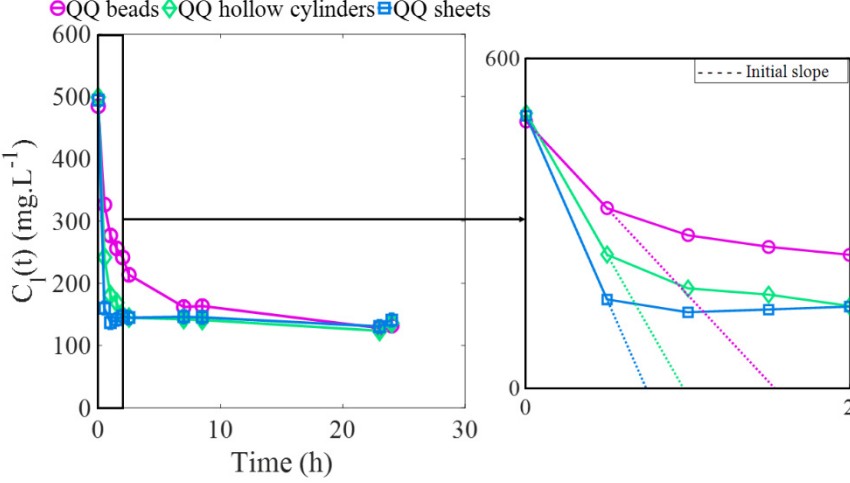

**Figure 5.** Concentration of the Rose Bengal lactone in the liquid phase, over time. Jar-test experiments.

**Table 4.** Linear correlation and parameters of the pseudo-second-order kinetics model for the adsorption of Rose Bengal lactone into solid media. Jar-test experiments.

|  | Beads | Hollow Cylinders | Sheets |
|---|---|---|---|
| Linear correlation parameters ($y = ax + b$) | | | |
| $a$ (g·mg$^{-1}$) | 0.3187 | 0.3157 | 0.3260 |
| $b$ (h·mg$^{-1}$·g) | 0.3015 | 0.0712 | 0.0376 |
| $R^2$ | 0.9996 | 0.9995 | 0.9996 |
| Model parameters | | | |
| $q_e$ (calculated) (mg·g$^{-1}$) | 3.14 | 3.17 | 3.07 |
| $q_e$ (experimental) (mg·g$^{-1}$) | 3.0 ± 0.2 | 3.1 ± 0.2 | 3.0 ± 0.2 |
| $k_2$ (g·mg$^{-1}$·h$^{-1}$) | 0.34 | 1.40 | 2.83 |
| $k_2$ ($10^{-5}$ g·mg$^{-1}$·s$^{-1}$) | 9.36 | 38.89 | 78.51 |
| $k_2 q_e$ ($10^{-4}$ s$^{-1}$) | 2.94 | 12.33 | 24.10 |

It appeared at first that the calculated maximal amounts, $q_e$, of Rose Bengal lactone that were adsorbed at equilibrium (also representing the adsorption capacities of the media), were very close to the experimental values reached after 24 h (Figure 4), with discrepancies of less than 5%, which confirms that the phenomenon studied had reached the stable state after 24 h, even for the beads. When comparing the final amounts, $q_e$, reached for the three shapes in each experiment, the differences were not significant, since the three (experimental) values differed by less than 5%. This finding indicates that the shape of the solid media has no effect on their adsorption capacities, which is consistent since the media were made from the same material. However, significant differences appeared in the pseudo-second-order rate constant of adsorption, $k_2$, (representing the adsorbed flux per unit of mass of the solid media), since it was found to be, respectively, 12 and 24 times greater for the hollow cylinders and sheets, than for the beads (Table 4). This trend was confirmed by the observation of the adsorption rate index ($q_e k_2$) defined in [35], which purely reflects the kinetic performance of the system. Thus, considering the same volumes of solid media, the adsorption kinetics performance for the sheets was considerably greater than for the hollow cylinders or the beads.

Knowing that the adsorption phenomenon involves a succession of mechanisms (external mass transfer, internal mass transfer, pore diffusion, etc.), and that the constant, $k_2$, englobes all of these mechanisms, these results prove that the differences between the three shapes of solid particles necessarily originate in at least one of the steps.

### 3.2. Analysis of the External Liquid–Solid Mass Transfer Mechanism

According to the method proposed by Furusawa and Smith [36], the external mass transfer step is related to the changes in concentrations over time, seen through the mass balance expressed in Equation (9), where $k_{ls}$ is the liquid–solid mass transfer coefficient (in m·s$^{-1}$); $S_s$ is the total area of the liquid–solid interface (in m$^2$); $V_l$ is the total liquid volume; and $C_l(t)$ and $C_s(t)$ are the concentration in the bulk and the concentration at the surface of the solid particles, respectively. A simple method was proposed to deduce $k_{ls}$, and it involved substituting Equation (9) with the corresponding values at the initial conditions (when $t \to 0$), which resulted in Equation (10). Therefore, the liquid–solid external mass transfer coefficient, $k_{ls}$, can be deduced from the initial slope $[(\frac{dC_l(t)}{dt})_{t=0}]$ of the curve, $C_l(t)$, versus time. This approach has been adopted in the literature to study different systems involving liquid–solid mass transfer, some examples of which are the removal of food dyes using chitosan particles [37,38] or the transfer of ions using ion-exchange resin [39].

$$\frac{V_l dC_l(t)}{dt} = -k_{ls}S_s(C_l(t) - C_s(t)) \tag{9}$$

$$\left(\frac{dC_l(t)}{dt}\right)_{t=0} = \frac{-k_{ls}S_s}{V_l}C_l(t=0) \tag{10}$$

This approach is based on the assumption that the concentration in the liquid, $C_l(t)$, is uniform. The evolution of the concentration in the liquid phase, $C_l(t)$, is presented in Figure 5.

In order to evaluate the external mass transfer coefficient in this study, the initial slope, $(\frac{dC_l(t)}{dt})_{t=0}$, of each curve was determined (Figure 5) and used to calculate the external mass transfer, $k_{ls}$, via Equation (10), which resulted in Equation (11). The results are presented in Table 5.

$$k_{ls} = -\frac{V_l}{S_s C_l(t_0)}\left(\frac{dC_l(t)}{dt}\right)_{t=0} \tag{11}$$

**Table 5.** External liquid–solid mass transfer parameters for the three media shapes and the three experiments, repeated for each shape. Jar-test experiments (under 90 rpm and solid:liquid ratio of 1:9).

| | Beads | Hollow Cylinders | Sheets |
|---|---|---|---|
| **Experiment 1** | | | |
| $S_s$ (m$^2$) | 0.100 | 0.135 | 0.234 |
| $Cl(t = 0)$ (mg·L$^{-1}$) | 484 | 499 | 495 |
| $(dCl(t)dt)t = 0$ (mg·L$^{-1}$·h$^{-1}$) | −316 | −514 | −668 |
| $k_{ls}S_s$ (10$^{-7}$ m$^3$·s$^{-1}$) | 0.95 | 1.51 | 1.98 |
| $k_{ls}$ (10$^{-6}$ m·s$^{-1}$) | 0.95 | 1.12 | 0.85 |
| **Experiment 2** | | | |
| $S_s$ (m$^2$) | 0.086 | 0.115 | 0.200 |
| $Cl(t = 0)$ (mg·L$^{-1}$) | 424 | 409 | 436 |
| $(dCl(t)dt)t = 0$ (mg·L$^{-1}$·h$^{-1}$) | −320 | −455 | −612 |
| $k_{ls}S_s$ (10$^{-7}$ m$^3$·s$^{-1}$) | 0.94 | 1.39 | 1.75 |
| $k_{ls}$ (10$^{-6}$ m·s$^{-1}$) | 1.12 | 1.21 | 0.88 |
| **Experiment 3** | | | |
| $S_s$ (m$^2$) | 0.086 | 0.115 | 0.200 |
| $Cl(t = 0)$ (mg·L$^{-1}$) | 506 | 492 | 503 |
| $(dCl(t)dt)t = 0$ (mg·L$^{-1}$·h$^{-1}$) | −300 | −456 | −639 |
| $k_{ls}S_s$ (10$^{-7}$ m$^3$·s$^{-1}$) | 1.14 | 1.74 | 2.38 |
| $k_{ls}$ (10$^{-6}$ m·s$^{-1}$) | 1.30 ± 0.1 | 1.51 ± 0.2 | 1.19 ± 0.1 |
| **Average values over the three experiments ± standard deviation** | | | |
| $k_{ls}S_s$ (10$^{-7}$ m$^3$·s$^{-1}$) | 1.0 ± 0.1 | 1.5 ± 0.2 | 2.0 ± 0.2 |
| $k_{ls}$ (10$^{-6}$ m·s$^{-1}$) | 1.1 ± 0.2 | 1.3 ± 0.2 | 1.0 ± 0.2 |
| $L$ (10$^{-3}$ m) | 1.75 | 0.45 | 0.25 |
| $D$ (10$^{-10}$ m$^2$·s$^{-1}$) | | 3.92 | |
| $Sh$ | 4.9 ± 0.4 | 1.5 ± 0.1 | 0.6 ± 0.08 |

The product, $k_{ls}S_s$, representing the transferred flux, describes the same trend as the adsorption kinetics parameters (greater for the sheets than for the hollow cylinders and beads). Despite the uncertainty in the product, $k_{ls}S_s$, for the different QQ shapes that can reach 30%, the differences between the three shapes were significant (on average 50% between the beads and cylinders, and 33% between the cylinders and sheets). This result confirmed the assumptions made in the adsorption analysis, that the shapes of the media induce significant differences in the external mass transfer step of the whole phenomenon. In particular, the external mass transfer is governed by the external surface area of the solid medium, which explains the greater flux obtained for the sheets. It should be pointed out that, in the case of the cylinder, the considered surface is the total one (sum of external, internal, and sides surfaces) and it is obvious that the liquid renewal inside the cylinder can be weak. Therefore, the measurement of the global transfer at the begining of the experiement could have been affected by a concentration of dye inside the QQ media that was lower than that measured in the liquid bulk. The external mass transfer coefficient, $k_{ls}$, was determined for the three shapes (Table 5). On average, an 18% difference was found between the mass transfer coefficients of the beads and the hollow cylinders, and a 30% difference was found between the hollow cylinders and sheets, which can be viewed as insignificant (<30%). Therefore, the external mass transfer coefficient for the three shapes of media can be considered as similar, which is consistent with the fact that the hydrodynamic conditions created in the jar tests were similar. Knowing the external liquid–solid mass transfer coefficient, as well as the diffusion coefficient of Rose Bengal in water, the Sherwood number could be determined for each condition, according to Equation (12).

$$Sh = \frac{k_{LS}L}{D} \tag{12}$$

In order to take account of the fact that the transfer occurs from all around a solid particle to its center, the characteristic length, *L*, in Equation (12) was taken to be the radius for the beads, and half the thickness for both the hollow cylinders and the sheets. According to the literature [38], the external convective mass transfer step is completely prevalent for *Sh* < 0.5, whereas the diffusion is significant for *Sh* > 10. In the present case (Table 5), the Sherwood numbers roughly ranged between 0.5 and 5, and, thus, were between 0.5 and 10, indicating that there was no strong predominance of one mechanism over another under the conditions investigated. This result showed that the internal diffusion and the external convection can be considered as two mechanisms of equivalent importance, both governing the overall transport of molecules from the bulk to the solid media. In addition, when the three shapes of the solid media particles were compared, the greatest Sherwood number was obtained for the solid beads, which means that the internal diffusion in this shape is more significant than for the other two shapes.

To conclude, from the perspective of applying quorum quenching for biofouling mitigation by using endoenzyme-producing bacteria (such as *Rhodococcus* sp. BH4), the (external) mass transfer of AHLs from the mixed liquor towards the core of the media could be favored, in terms of the transferred flux ($k_{ls}S_s$), by the presence of sheets, rather than the other two shapes, for the same volume of medium. The beads seem to be the least favorable shape for mass transfer: not only does their small surface area give rise to the lowest transferred flux ($k_{ls}S_s$) (in comparison to the other shapes), but their great diameter also slows down the internal transfer step. Considering that quorum quenching is based on an enzymatic reaction, further investigation is needed to know whether the transferred flux or the transfer kinetics of AHLs (substrate) towards enzymes is the key parameter to promote, when the aim is to reduce biofilm formation effectively.

## 4. Solid–Liquid Mass Transfer from the Solid Media to the Liquid

In the case of the experiments conducted in the aerated lab-scale ALMBR, the transfer of the Rose Bengal lactone took place from the solid media to the liquid. Different operating conditions were investigated to highlight the effect of the solid medium shape (beads, hollow cylinders, and flat sheets), as well as the effect of the hydrodynamics (the three air flowrates and the specific aeration demand (SAD), of 0.75, 0.9, and 1 $\text{Nm}^3 \cdot \text{h}^{-1} \cdot \text{m}^{-2}$) on the transfer from the solid media phase to the liquid phase. This type of configuration (solid–liquid) has been much less investigated than the previous one (liquid–solid), especially under similar hydrodynamic conditions. Although it can be assumed that the transfer from the solid to the liquid is a combination of several mechanisms, as illustrated in Figure 6, to the best of our knowledge, no complete model of the whole phenomenon, taking all the steps involved into account, has been reported in the literature.

Some studies have focused on characterizing the solid–liquid mass transfer involved in the dissolution of solid substances in their own solution, using the mass balance presented in Equation (13) [40]. This approach, which is similar to the one developed for the liquid–solid mass transfer (the opposite phenomenon) (Equation (9)), allows a coefficient, $k_{sl}$, to be determined for the solid–liquid external mass transfer. The solid–liquid mass transfer coefficient can be deduced from Equation (14), which corresponds to the very early stage of the operation (when $t \to 0$), at which the concentration of the liquid is equal to zero and the initial concentration at the surface of the solid, $C_s(t = 0)$, can be considered as homogeneous in the solid media, and equals the ratio of the initial mass of the dye ($m_0$) adsorbed by the QQ media, to the total volume of solid media, $V_s$. In practice, $k_{sl}$ is deduced from the initial slope of the curve, $C_l(t)$, versus time.

$$\frac{V_l dC_l(t)}{dt} = k_{sl}S_s(C_s(t) - C_l(t)) \tag{13}$$

$$\left(\frac{dC_l(t)}{dt}\right)_{t=0} = \frac{k_{sl}S_s}{V_l}C_s(t=0) = \frac{k_{sl}S_s m_0}{V_l V_s} \tag{14}$$

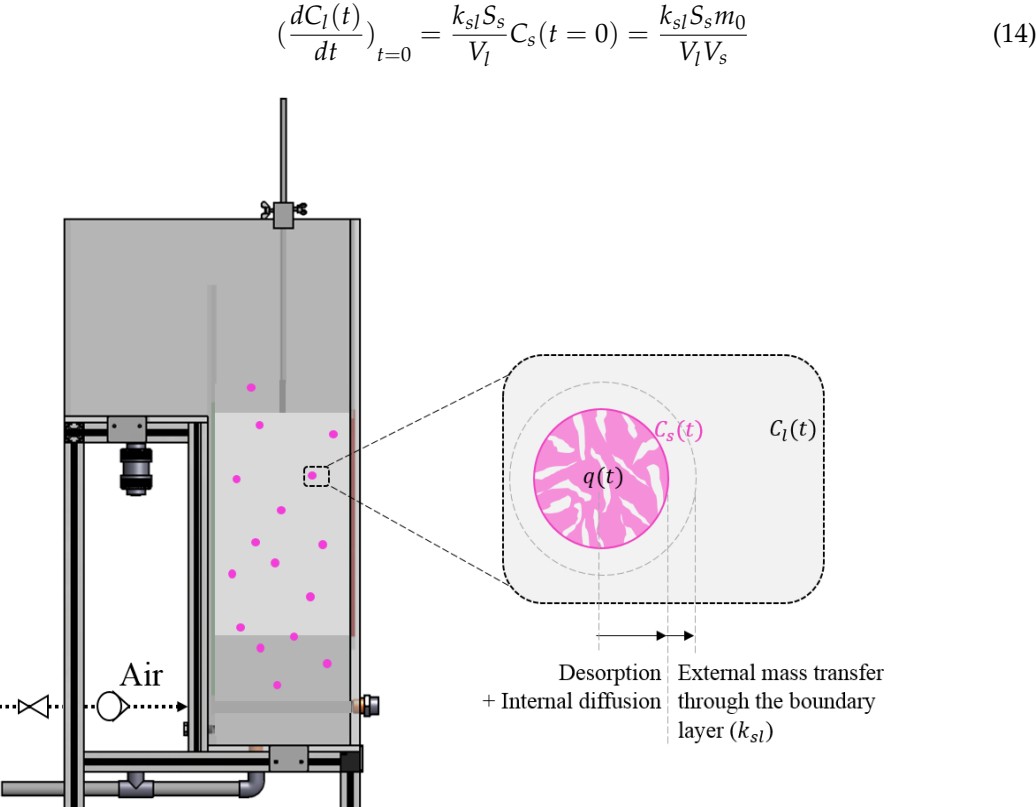

**Figure 6.** Illustrative representation of the transfer of Rose Bengal lactone from the solid medium to the liquid, in the aerobic tank of the lab-scale MBR.

The mass transfer from the media to the liquid phase was studied in the ALMBR under different aeration conditions, in order to evaluate how the hydrodynamics and the QQ media shape could influence the transfer. However, before analyzing the results, it is worth mentioning that, in the case of this study, it can be assumed that the hydrodynamics were involved at several levels in the mass transfer phenomenon, as follows: by promoting turbulence in the liquid phase, it can reduce the concentration gradient between the surface of the media and the bulk and, thus, accelerate the mass transfer phenomenon; at the local scale, it can enhance the motion of the particles, and, therefore, favor the fast renewal of the liquid boundary layer around them. In addition the air flowrate affects the proportion of fluidized QQ media in the reactor, and, thus, it can control the exchange surface between the solid media and the liquid phase. Therefore, it can be assumed that the effect of hydrodynamics on the global mass transfer is complex because of the coupling of all these phenomena. For each air flowrate, the total surface corresponding to the added QQ media, S, the surface corresponding to the moving particles (fluidized ones), $S_{FM}$, and the volume of these suspended QQ media, $V_{FM}$, are reported in Table 6. Since the used QQ media were those obtained at the end of the jar tests, their initial dye concentration was known.

**Table 6.** Solid media properties in the ALMBR, under different air flowrates.

| SADm ($Nm^3 \cdot h^{-1} \cdot m^{-2}$) | 0.75 | | | 0.90 | | | 1.00 | | |
|---|---|---|---|---|---|---|---|---|---|
| Solid medium | Beads | Hollow cylinders | Sheets | Beads | Hollow cylinders | Sheets | Beads | Hollow cylinders | Sheets |
| Total exchange surface area, $S_s$ ($m^2$) | 0.086 | 0.115 | 0.200 | 0.086 | 0.115 | 0.200 | 0.086 | 0.135 | 0.234 |
| Fluidization rate (%) | 6.2 | 26.2 | 30.1 | 10.9 | 53.2 | 47.8 | 16.1 | 63.3 | 55.9 |
| Fluidized surface area, $S_{FM}$ ($m^2$) | 0.0062 | 0.0354 | 0.0704 | 0.0109 | 0.0718 | 0.1118 | 0.0161 | 0.0845 | 0.1308 |
| Fluidized volume, $V_{FM}$ ($10^{-5}$ $m^3$) | 0.31 | 1.53 | 1.76 | 0.64 | 3.11 | 2.79 | 0.94 | 3.70 | 3.27 |

*4.1. Effect of Hydrodynamics on Solid–Liquid Mass Transfer*

Using the camera and the image processing, the concentration of Rose Bengal lactone over time was monitored when the stained QQ media were introduced into the ALMBR, under the conditions presented in Table 6. The results are presented in Figure 7, in terms of the normalized amount of dye, ($\frac{C_l(t)V_l}{m_0}$), released into the liquid over time. It is first possible to observe that the overall aspect of the curves includes a rapid increase at the beginning and then tends toward a stable value. This type of trend was to be expected since, in this case, the mass transfer was driven by the difference in the concentrations between the solid and the liquid, which tends to be attenuated over time, inducing a slowdown in the transfer. The overall mass transfer phenomenon appeared to be relatively slow, since all the experiments lasted more than 24 h. The time required to approach the equilibrium (the moment, $t_e$, at which the normalized concentration reached 96% of its final value) was measured for each condition. The times, $t_e$, for the three different air flowrates were quite similar for a given shape of solid media. It could then be assumed that the global transfer dynamics were not significantly affected by the aeration in the range of air flowrates investigated in this study.

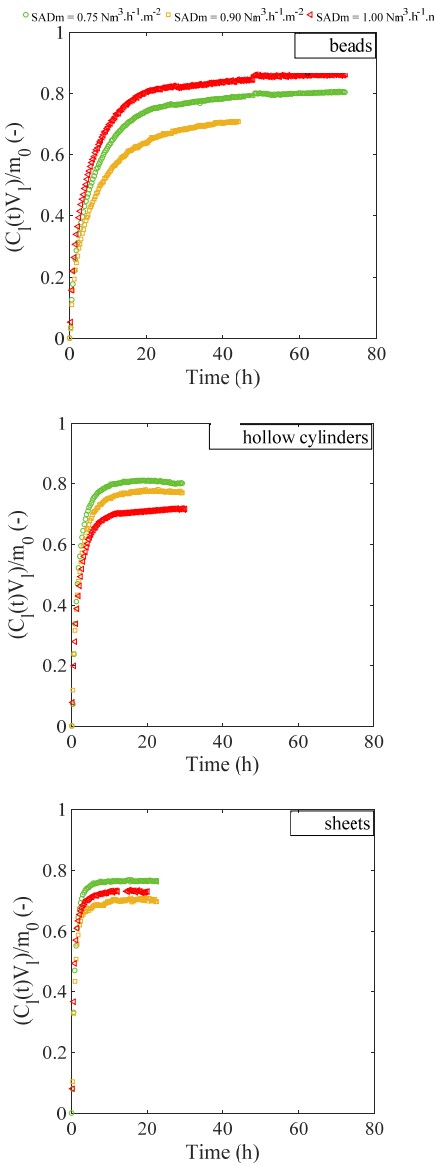

**Figure 7.** Effect of the hydrodynamics on the normalized amount of Rose Bengal lactone released into the liquid phase for the three different shapes of solid media. ALMBR experiments.

In addition, the solid–liquid mass transfer parameters ($k_{sl}$ and $k_{sl}S_{exchange}$, where $S_{exchange}$ is the exchange surface between the solid medium and the liquid phase) were determined considering the initial conditions ($t \to 0$) and according to the approach explained previously (Equations (13) and (14)). The initial slopes of the curves, $C_l(t)$, were determined over the first 10 points (the first 90 s of the operation) for each condition, and one example is presented in Figure 8 for the mass transfer from the beads to the liquid, under the three air flowrates investigated in this study.

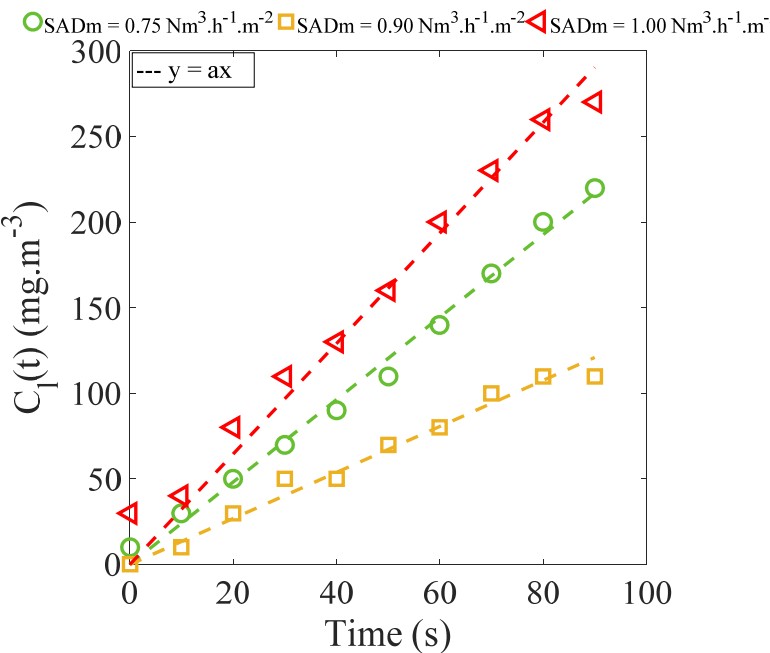

**Figure 8.** Initial evolution (over the first 90 s) of the concentration of Rose Bengal lactone in the liquid phase, using beads with three air flowrates. ALMBR experiments.

The correlation parameters and the mass transfer parameters are grouped in Table 7 by the mass transfer parameters obtained for the different shapes. The observation of the transferred flux, $k_{sl}S_{exchange}$, revealed no clear effect of the aeration in the range investigated, since different trends are described for the three shapes of solid media. The determination of the mass transfer coefficient, $k_{sl}$, requires the transferred flux that was measured to be divided by the exchange surface area, $S_{exchange}$, to compare the transfer kinetics. However, in the present case, the exact exchange surface area was hard to evaluate, since it depended on the fluidization of the solid media, as previously mentioned. Therefore, two possible limit-cases could be considered for discussion. In the first case, the exchange surface area corresponds to the total surface, $S_s$, of solid medium introduced into the reactor, whether the particles are fluidized or settled. In the second case, the exchange surface corresponds to the surface area, $S_{FM}$, of the fluidized particles only. The mass transfer coefficients were calculated in both cases (Table 7) and are also represented as a function of the aeration for the three shapes of solid media in Figure 9.

Significant differences can be observed between the two cases considered. In the first case, where the totality of the solid media introduced (with surface area, $S_s$) into the reactor was assumed to participate in the global transfer, the mass transfer coefficients ranged roughly between $0.5 \times 10^{-7}$ and $2.0 \times 10^{-7}$ m·s$^{-1}$. In the second case, where the assumption was that only the fluidized particles (with surface area, $S_{FM}$) participate in the global transfer, the mass transfer coefficients ranged between $0.2 \times 10^{-6}$ and $2.0 \times 10^{-6}$ m·s$^{-1}$. These observations were consistent and actually indicate that, in the second case, where the number of particles transferring (only the fluidized particles) is lower, the mass transfer would have to be much faster to reach the same transferred flux as in the first case. However,

the effect of the air flowrate on the mass transfer coefficient was still hardly identifiable in the investigated range, whichever case is considered.

These results show that the effect of the air flowrate on the solid–liquid mass transfer was not clearly identifiable in the conditions investigated because of the complexity of the mechanisms, which depend on the aeration and on the difficulty of assessing the exchange surface area involved in the transfer with certainty. The two cases considered here are "extreme" cases, which are most likely not representative of the reality since, (i) the settled media can also take part in the global transfer, and (ii) because there is an alternation phenomenon in the fluidization (the media in suspension are not necessarily the same during the whole operation). Thus, the effective surface area of exchange, $S_{exchange}$, between the solid medium and the liquid phase should be somewhere between the surface area of the fluidized particles and the total surface area of the medium ($S_{FM} < S_{exchange} < S_s$).

**Table 7.** Parameters of the mass model and Sherwood numbers for the solid–liquid mass transfer of Rose Bengal lactone from the medium to the liquid, under different air flowrates (The *Sh* numbers were calculated with the diffusion coefficient: $D = 3.92 \times 10^{-10}$ m$^2 \cdot$s$^{-1}$). ALMBR experiments.

| SADm (Nm$^3 \cdot$h$^{-1} \cdot$m$^{-2}$) | 0.75 | 0.90 | 1.00 |
|---|---|---|---|
| Solid–liquid mass transfer parameters for beads | | | |
| Total exchange surface area, $S_s$ (m$^2$) | 0.086 | 0.086 | 0.086 |
| Fluidization rate (%) | 6.2 | 10.9 | 16.1 |
| Fluidized surface area, $S_{FM}$ (m$^2$) | 0.062 | 0.0109 | 0.01611 |
| Fluidized volume, $V_{FM}$ (10$^{-5}$ m$^3$) | 0.31 | 0.64 | 0.94 |
| $k_{sl}S_{exchange}$ (10$^{-8}$ m$^3 \cdot$s$^{-1}$) | 1.00 | 0.62 | 1.52 |
| $k_{sl}$ (considering $S_s$) (10$^{-7}$ m$\cdot$s$^{-1}$) | 1.17 ± 0.1 | 0.72 ± 0.06 | 1.78 ± 0.15 |
| $k_{sl}$ (considering $S_{FM}$) (10$^{-7}$ m$\cdot$s$^{-1}$) | 18.8 ± 1 | 6.8 ± 0.5 | 11 ± 1 |
| $L$ (10$^{-3}$ m) | | 3.50 | |
| $Sh$ (considering $S_s$) (-) | 1.04 | 0.66 | 1.59 |
| $Sh$ (considering $S_{FM}$) (-) | 16.80 | 6.08 | 9.86 |
| Solid–liquid mass transfer parameters for hollow cylinders | | | |
| Total exchange surface area, $S_s$ (m$^2$) | 0.115 | 0.115 | 0.135 |
| Fluidization rate (%) | 26.2 | 53.2 | 63.3 |
| Fluidized surface area, $S_{FM}$ (m$^2$) | 0.0354 | 0.0718 | 0.0845 |
| Fluidized volume, $V_{FM}$ (10$^{-5}$ m$^3$) | 1.53 | 3.11 | 3.70 |
| $k_{sl}S_{exchange}$ (10$^{-8}$ m$^3 \cdot$s$^{-1}$) | 2.16 | 1.70 | 1.87 |
| $k_{sl}$ (considering $S_s$) (10$^{-7}$ m$\cdot$s$^{-1}$) | 1.88 ± 0.1 | 1.48 ± 0.1 | 1.39 ± 0.1 |
| $k_{sl}$ (considering $S_{FM}$) (10$^{-7}$ m$\cdot$s$^{-1}$) | 7.16 ± 0.6 | 2.86 ± 0.3 | 2.20 ± 0.2 |
| $L$ (10$^{-3}$ m) | | 0.90 | |
| $Sh$ (considering $S_s$) (-) | 0.43 | 0.35 | 0.32 |
| $Sh$ (considering $S_{FM}$) (-) | 1.65 | 0.66 | 0.50 |
| Solid-liquid mass transfer parameters for sheets | | | |
| Total exchange surface area, $S_s$ (m$^2$) | 0.200 | 0.200 | 0.234 |
| Fluidization rate (%) | 30.1 | 47.8 | 55.9 |
| Fluidized surface area, $S_{FM}$ (m$^2$) | 0.0704 | 0.01118 | 0.1308 |
| Fluidized volume, $V_{FM}$ (10$^{-5}$ m$^3$) | 1.76 | 2.79 | 3.27 |
| $k_{sl}S_{exchange}$ (10$^{-8}$ m$^3 \cdot$s$^{-1}$) | 4.03 | 3.23 | 4.58 |
| $k_{sl}$ (considering $S_s$) (10$^{-7}$ m$\cdot$s$^{-1}$) | 2.02 ± 0.2 | 1.62 ± 0.1 | 1.95 ± 0.2 |
| $k_{sl}$ (considering $S_{FM}$) (10$^{-7}$ m$\cdot$s$^{-1}$) | 6.63 ± 0.5 | 3.37 ± 0.3 | 3.50 ± 0.3 |
| $L$ (10$^{-3}$ m) | | 0.50 | |
| $Sh$ (considering $S_s$) (-) | 0.26 | 0.21 | 0.25 |
| $Sh$ (considering $S_{FM}$) (-) | 0.85 | 0.43 | 0.45 |

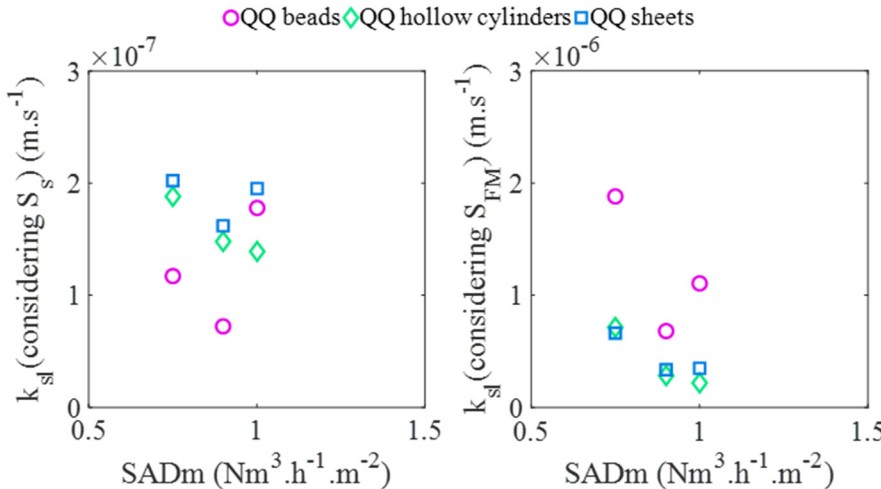

**Figure 9.** Mass transfer coefficients under different air flowrates and for the three shapes of media particles. ALMBR experiments.

### 4.2. Effect of Solid Particle Shapes on the Mass Transfer

Because the initial conditions were different from one experiment to another and the levels reached at equilibrium were different for the three shapes of solid particles (Figure 7), the concentration curves were normalized with respect to the final value, $\left( \frac{C_l(t)}{C_l(t_\infty)} \right)$, and the curves presented in Figure 10 only take the dynamics of the transfer phenomenon into account (independently of the amounts transferred).

The effect of the media shapes can be determined, first, by visualizing the curves, since it appears clearly that the overall dynamics were faster for the sheets than for the hollow cylinders or the beads. This can be confirmed by the comparison of the times, $t_e$, necessary to reach 96% of the equilibrium. It appears that the time to reach equilibrium for the beads was approximately three times greater than that for the hollow cylinders, and nine times that for the sheets, in the range of air flowrates investigated. In the case of the beads, the curves presented in Figure 10 reveal that the equilibrium was not completely reached and that the mass transfer continued to evolve very slowly, possibly because of their larger diameter and, thus, their slower internal mass transfer. The mass transfer parameters, determined in Table 7, show that significant differences were obtained in the transferred flux, $k_{sl}S_{exchange}$, for the three shapes. For example, for a SADm of 1.00 Nm$^3$.h$^{-1}$.m$^{-2}$ (corresponding to the highest fluidization rates), $k_{sl}S_{exchange}$, in the presence of the sheets was up to three or four times greater than that of the beads or the hollow cylinders, respectively. In terms of the mass transfer coefficient, $k_{sl}$, the differences between the three particle shapes depended on which of the two cases mentioned above was considered. It appears that, for the same volume of solid medium introduced into the reactor, the mass transfer phenomenon was considerably enhanced in the presence of the sheets, compared to the other shapes, in terms of both the time taken to reach equilibrium and the flux transferred. In addition, the time, $t_e$, needed to reach equilibrium for the sheets was only approximately 4 h, which represents one third of the residence time (12 h) of the ALMBR, whereas the time needed for the beads (approximately 30 h) was at least 2.5 times greater than the residence time. These results suggest that, among the three shapes of media, the sheets would be the most favorable to enhance the mass transfer phenomenon from the medium to the liquid phase, and, thus, to quickly mitigate fouling via the biological activity. The Sherwood numbers were determined according to Equation (12) for each medium shape and are shown in Table 7. Again, two cases were considered, corresponding to the two exchange surface areas. When the total surface, $S_s$, was taken into account, the Sherwood numbers ranged from 0.2 to 1.6 for the different particle shapes, under the different air flowrates. According to the literature, the fact that the Sherwood numbers were lower than 0.5 (in this case for the hollow cylinders and the sheets) indicates that the diffusion

mechanism was very fast, compared to the external mass transfer step in the investigated conditions, and that the diffusion time can be considered as negligible. However, for the beads ($0.6 < Sh < 1.6$), the internal diffusion was not negligible, which is consistent with the fact that equilibrium was never completely reached in the investigated conditions. For $Sh <1$, it was possible to propose a scaling law between the Sherwood number and the media Reynolds number, ($Re = \frac{\rho_{fluid} \times U_{liquid\ in\ the\ riser}\ d_{media}}{\mu_{fluid}}$), with good precision (22% of standard deviation), as follows:

$$Sh = 0.1 \times Re^{0.2} \tag{15}$$

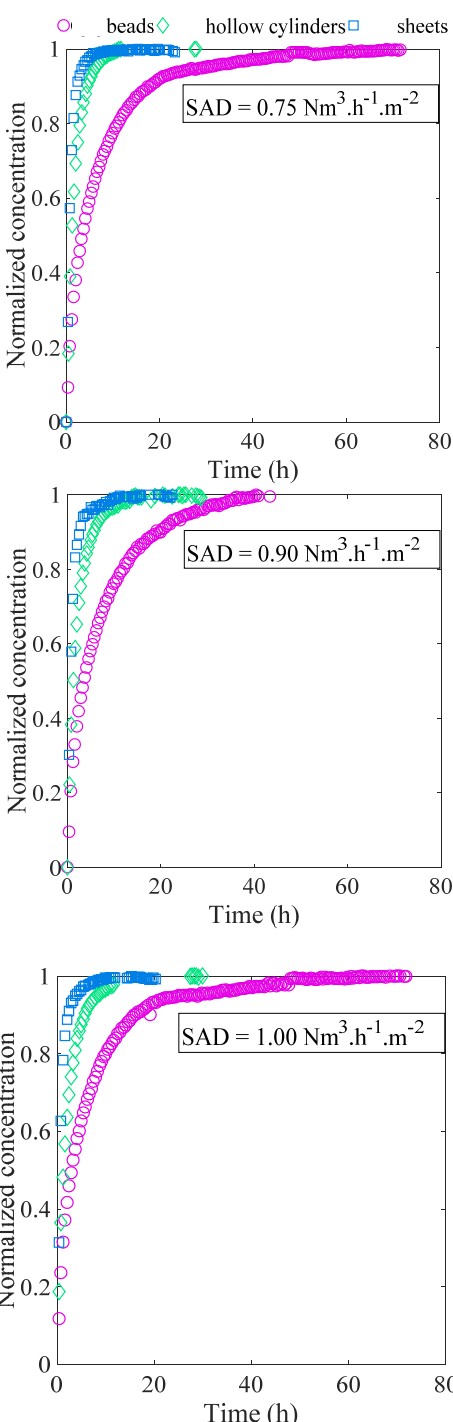

**Figure 10.** Effect of the shapes of the solid media on the normalized concentrations of Rose Bengal lactone in the liquid phase, for the three different air flowrates. ALMBR experiments.

This correlation shows the strong effect of the local hydrodynamics, impacted by the particle on the global mass transfer rate of the medium for all the shapes studied. It is worth noting that this correlation could be improved with more data, to include other dimensionless numbers, such as the Schmidt number. When the fluidized surface area, $S_{FM}$, was considered, the Sherwood numbers were much greater and ranged from 0.4 to 17. In most cases, the Sherwood numbers were between 0.5 and 10 (except for the beads, under 0.75 $Nm^3 \cdot h^{-1} \cdot m^{-2}$), which, according to the literature, indicates that there was no clear prevalence of one mechanism over another. However, in the particular case of the beads under a SADm of 0.75 $Nm^3 \cdot h^{-1} \cdot m^{-2}$, the Sherwood number was 16.80 (>10), which refers to a slow diffusion step. In all cases (whatever the exchange surface considered), the Sherwood numbers obtained for the beads were much higher than for the other two shapes, which, undeniably, shows that the diffusion into the beads was slower because of their larger diameter.

A comparison of the investigated air flowrates led to the conclusion that no notable effect on the solid–liquid mass transfer coefficient was observed in the conditions of the study. This result could be explained by the fluctuations in the fluidization of the media during the experiment and the difficulty of evaluating the effective exchange surface with precision, which made the precise determination of a mass transfer coefficient difficult. Concerning the effect of the media shapes, it was found that the sheets gave rise to greater mass transfer, in the shortest time, to reach equilibrium and the greatest transferred flux. With a view to applying quorum quenching to reduce membrane biofouling, sheets may, therefore, be the most appropriate shape in the case of exoenzyme-producing bacteria (*Pseudomonas* sp. 1A1, for example), as the exoenzyme produced in the core of a sheet could rapidly transfer out of it to reach and degrade the AHLs present in the mixed liquor.

Finally, the determination of the Sherwood number in the conditions investigated demonstrated that the overall phenomenon is clearly limited more by the external mass transfer step, than by the internal diffusion in the solid media. Nevertheless, comparing the three shapes of media tended to show that the diffusion could, again, be a little more important in the case of the beads, as was the case for the liquid–solid mass transfer, because of their greater diameter.

In order to evaluate the impact of the presence of solid media on a possible membrane clogging, the permeability of the membrane was measured for several operating conditions. In the case of a constant flow filtration, clogging is characterized by the increase in transmembrane pressure (TMP) over time. A pressure sensor on the permeate side allowed us to follow the pressure $Pp$ over time. The initial pressure, $P_i$, in the reactor was measured at the beginning of the experiment, when the pump on the permeate side was stopped. The transmembrane pressure is calculated according to the equation below:

$$\text{TMP} = P_i - Pp \tag{16}$$

The permeability measurement was performed by maintaining a constant volume of water in the reactor. The operation consisted of setting the permeate flow rate and following the pressure evolution to determine the permeate flux, $J$, and the permeability, $Lp$, and was determined according to Darcy's law, with the following expression:

$$J_{20\,°C} = Lp \times TMP = \frac{PTM}{\mu_{20\,°C} \times R_m} \tag{17}$$

Figure 11 represents the evolution of the water permeability at 20 °C, as a function of time. It was obtained for a SADm of 0.75 $Nm^3/h/m^2$. The first experiment was carried out with demineralized water, without the addition of media, it will be used as reference. Two other experiments were performed with the addition of inert media (beads and hollow cylinders) for a volume fraction of 0.45% $v/v$. The results highlighted the establishment of an operating regime that stabilizes after one hour of filtration. If we consider only the measurements from one hour of filtration, in which the permeability measurement

was stable, for the three experiments carried out (in the presence or absence of media), under similar operating conditions, it can be seen that the results of the water filtration were practically identical (considering an error in the permeability measurement of ±15%). It can, therefore, be stated that the addition of inert media does not lead to membrane clogging, since the permeability is similar with or without QQ media.

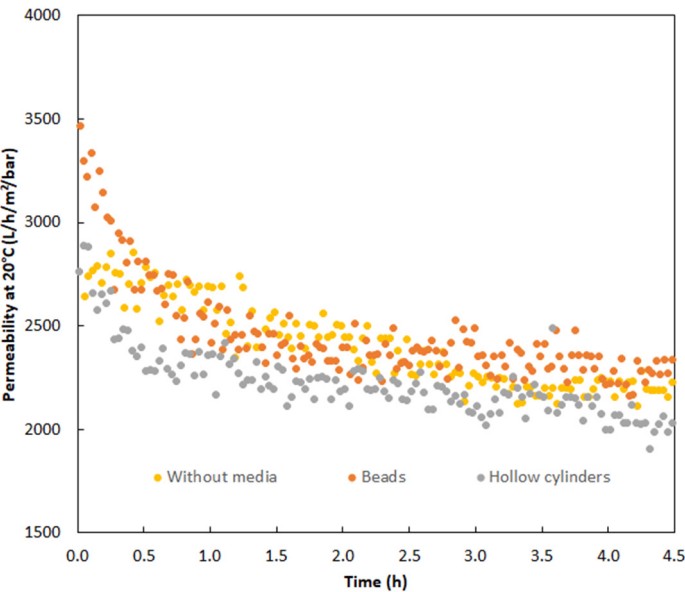

**Figure 11.** Evolution of the permeability of the membrane for water at 20 °C, with and without solid media.

## 5. Conclusions

In this article, an attempt has been made to quantitatively characterize the mass transfer phenomena that can be involved in the application of a bacterial antifouling technique (quorum quenching), using different shapes of alginate supports for bacteria, for the first time in the literature on this topic. Considering that the quorum quenching can be based on an endoenzyme-producing bacterium or an exoenzyme-producing one, two types of mass transfer were distinguished for the study: liquid–solid mass transfer, to mimic the transport of signal molecules from the mixed liquor to the inner part of the solid media; and solid–liquid mass transfer, to mimic the transport of exoenzymes from the solid media to the mixed liquor. Using a model (dye) molecule under different hydrodynamic conditions, the mass transfer kinetics characterization enabled the following main conclusions to be drawn. First, the tools selected for the characterization of the mass transfer proved to be efficient for the purpose. The experimental setups and the theoretical approaches provided relevant information about both kinds of mass transfer: the study of the liquid–solid mass transfer in the jar tests determined the mass transfer coefficients in the order of magnitude of $10^{-6}$ m·s$^{-1}$. The results brought to light the fact that both the sheets and the hollow cylinders are appropriate shapes for an efficient mass transfer of AHLs from the mixed liquor to the entrapped endoenzyme-producing bacteria, in terms of the transferred flux, under similar hydrodynamic conditions. For the beads, the liquid–solid mass transfer was found to be less efficient because their specific shape gave the internal diffusion step more weight in the overall transfer phenomenon. The study of the solid–liquid mass transfer in the aerated ALMBR was performed under different air flowrates for the three shapes of media. The effect of the air flowrate on the mass transfer coefficient was hardly quantifiable because of the difficulty of assessing the effective exchange surface area. However, the effect of the shape could be determined and the most favorable shape, in terms of the transferred flux, appears to be the sheet. In the case of exoenzyme-producing bacteria entrapped in the solid media, the transfer of exoenzymes can be more efficient for sheets than for hollow cylinders or beads. The investigation of the two types of mass transfer gave interesting

insights into quorum quenching's application. Actually, the mass transfer coefficient for the liquid–solid transfer (in the jar test) was found to be of the order of magnitude of $10^{-6}$ m·s$^{-1}$, whereas the solid–liquid transfer (in the aerated ALMBR) was $10^{-7}$ m·s$^{-1}$. The two mass transfer coefficients are not directly comparable because they were obtained under different hydrodynamic conditions; however, this indicates that mechanical stirring can provide better hydrodynamic conditions to foster mass transfer. In all the cases, the beads were found to give rise to the smallest transferred flux, compared to the other two shapes. However, when the fluidized surface area, $S_{FM}$, was considered, the mass transfer coefficient for the beads was greater, which means that this shape could be valuable in that case, by increasing the surface (decreasing the diameter) and increasing their fluidization (increasing the air flowrate and decreasing their diameter and/or density). These findings are of great importance because they open the way to an optimized application of quorum quenching. Finally, the development of the optical technique was revealed to be relevant to the study of the solid–liquid mass transfer from the solid media to the liquid phase of the aerated part of the ALMBR.

**Author Contributions:** Conceptualization, N.B., M.M., C.L.M., J.T., C.L., N.D., C.-H.L. and C.G.; methodology, N.B., M.M., C.L.M., J.T., C.L., N.D., C.-H.L. and C.G.; software, N.B., M.M. and C.L.M.; validation, N.B., M.M., C.L.M., J.T., C.L., N.D., C.-H.L. and C.G.; formal analysis, N.B., M.M., C.L.M., J.T., C.L., N.D., C.-H.L. and C.G.; investigation, N.B., M.M., C.L.M., J.T., C.L., N.D., C.-H.L. and C.G.; resources, N.B., M.M., C.L.M., J.T., C.L., N.D., C.-H.L. and C.G.; data curation, N.B., M.M. and C.L.M.; writing—original draft preparation, N.B., M.M., J.T., C.L., N.D. and C.G.; writing—review and editing, N.B., M.M., C.L.M., J.T., C.L., N.D., C.-H.L. and C.G.; visualization, N.B., C.L.M. and N.D.; supervision, J.T., C.L., N.D. and C.G.; project administration, J.T., C.L., N.D. and C.G.; funding acquisition, C.L., N.D. and C.G. All authors have read and agreed to the published version of the manuscript.

**Funding:** This research received no external funding.

**Data Availability Statement:** Not applicable.

**Conflicts of Interest:** The authors declare no conflict of interest.

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
