# Peer review of "Experimental Solid–Liquid Mass Transfer around Free-Moving Particles in an Air-Lift Membrane Bioreactor with Optical Techniques"

_fluids, doi:10.3390/fluids7100338_

Round 1

Reviewer 1 Report

The manuscript reports the study of the mass transfer involved in the application of a bacterial antifouling technique for Membrane Bioreactors (MBR) via the addition of solid media. The transfer from liquid to "solid media" and the transfer from "solid media" to liquid phase are distinguished and quantified. The effect of aeration in the MBR was investigated and an optimal air flow rate to foster the transfer was found, based on the highest transfer coefficient that was obtained. Overall, the manuscript is well written, and the structure is well organized. The language is satisfactory, and the findings are meaningful. Thus, I am glad to recommend the publication of the present manuscript before the reviewer address the following comments.

1.     In lines 42-46 on page 2, the authors pointed out that the formation of the fouling layer on the member surface plays a key role in wastewater treatment in MBRs. A recent publication regarding the powder transportation mechanisms on membranes may help support the viewpoint (https://doi.org/10.1016/j.seppur.2022.122076).

2.     In Figure 1 and Table 1, what is the criterion for selecting the morphology of the solid media? Do the three solid media have the same volume?

3.     A typo in line 125 on page 4, “…are summarized by”. A reference may be omitted.

4.     In line 270 on page 8, “Another model, the pseudo-second-order model,”, the related reference for this model should be added here.

5.     In line 278 on page 8, “solid particulates” should be amended as “solid particles”.

6.     In lines 330-340 on page 10, the effects of particle morphology on the mass transfer performance are presented but the underlying mechanism lacks.

7.     In line 418 on page 13, “According to the literature,”. The related reference should be added.

8.     In Fig. 11 on page 23, the range of the Y-axis can be ranged from 1500 to 4000 to show the difference in the three solid media.

Author Response

We are extremely grateful for the suggestions of improvement that were realized in this revision. We have modified our paper by taking into account the comments of the reviewer. A special response for reviewer has been proposed in this document.

“1.     In lines 42-46 on page 2, the authors pointed out that the formation of the fouling layer on the member surface plays a key role in wastewater treatment in MBRs. A recent publication regarding the powder transportation mechanisms on membranes may help support the viewpoint (https://doi.org/10.1016/j.seppur.2022.122076).”

  • The reference has been added

“2.     In Figure 1 and Table 1, what is the criterion for selecting the morphology of the solid media?”

  • The selection was based on morphology the most used in literature.

“Do the three solid media have the same volume?”

  • As stated in line 168 “fraction of 0.45 % v/v with respected to the total volume of reactor (13 L).”, then the three solid media have the same volume fraction

“3.     A typo in line 125 on page 4, “…are summarized by”. A reference may be omitted.”

  • The reference has been added

“4.     In line 270 on page 8, “Another model, the pseudo-second-order model,”, the related reference for this model should be added here.”

  • A reference has been added

“5.     In line 278 on page 8, “solid particulates” should be amended as “solid particles”.”

  • It has been corrected
  1. In lines 330-340 on page 10, the effects of particle morphology on the mass transfer performance are presented but the underlying mechanism lacks.
  • A reference has been added
  1. In line 418 on page 13, “According to the literature,”. The related reference should be added.
  • A reference has been added
  • A reference has been added
  1. In Fig. 11 on page 23, the range of the Y-axis can be ranged from 1500 to 4000 to show the difference in the three solid media.
  • The figure has been changed

Reviewer 2 Report

The authors presented a very interesting experimental work on the solid-liquid mass transfer around free-moving particles in an air lift membrane bioreactor with optical techniques.

The paper can be accepted after minor revision:

The conclusion is to be extended.

The novelty of the work is to be clearly stated.

An actual photo of the experimental setup is to be presented.

Why no 2D visualizations are presented?

An experimental uncertainty study is to be performed.

The paper is to be checked against misprints and grammatical mistakes.

Author Response

“The authors presented a very interesting experimental work on the solid-liquid mass transfer around free-moving particles in an air lift membrane bioreactor with optical techniques. The paper can be accepted after minor revision:”

“The conclusion is to be extended/The novelty of the work is to be clearly stated.”

  • The conclusion has been improved underlying the novelty when possible.

An actual photo of the experimental setup is to be presented.

  • We do not have an actual photo of the running experiments

Why no 2D visualizations are presented?

  • The 2D visualization have been done but are not relevant as the concentration map in each picture do not provide supplementary information

An experimental uncertainty study is to be performed.

  • Precision has been added in table 7.

The paper is to be checked against misprints and grammatical mistakes.

  • The paper has been checked and corrected by a native English speaker